# Revisiting the *Tigger* Transposon Evolution Revealing Extensive Involvement in the Shaping of Mammal Genomes

**DOI:** 10.3390/biology11060921

**Published:** 2022-06-16

**Authors:** Mohamed Diaby, Zhongxia Guan, Shasha Shi, Yatong Sang, Saisai Wang, Yali Wang, Wencheng Zong, Numan Ullah, Bo Gao, Chengyi Song

**Affiliations:** 1College of Animal Science & Technology, Yangzhou University, Yangzhou 225009, China; dh18035@yzu.edu.cn (M.D.); mx120190652@yzu.edu.cn (Z.G.); mx120190668@yzu.edu.cn (S.S.); dx120170084@yzu.edu.cn (S.W.); dx120180099@yzu.edu.cn (Y.W.); numanhashmi@aup.edu.pk (N.U.); bgao@yzu.edu.cn (B.G.); 2School of Life Sciences, Sun Yat-sen University, Guangzhou 510275, China; sangyt@mail2.sysu.edu.cn; 3Institute of Animal Sciences, Chinese Academy of Agricultural Sciences, Beijing 100193, China; 82101211207@caas.cn

**Keywords:** transposons, *pogo*, *Tigger*, evolution, horizontal transfer

## Abstract

**Simple Summary:**

Despite the discovery of the *Tigger* family of pogo transposons in the mammalian genome, the evolution profile of this family is still incomplete. Here, we conducted a systematic evolution analysis for *Tigger* in nature. The data revealed that *Tigger* was found in a broad variety of animals, and extensive invasion of *Tigger* was observed in mammal genomes. Common horizontal transfer events of *Tigger* elements were observed across different lineages of animals, including mammals, that may have led to their widespread distribution, while parasites and invasive species may have promoted *Tigger* HT events. Our results also indicate that the activity of *Tigger* transposons tends to be low in vertebrates; only one mammalian genome and fish genome may harbor active *Tigger*.

**Abstract:**

The data of this study revealed that *Tigger* was found in a wide variety of animal genomes, including 180 species from 36 orders of invertebrates and 145 species from 29 orders of vertebrates. An extensive invasion of *Tigger* was observed in mammals, with a high copy number. Almost 61% of those species contain more than 50 copies of *Tigger*; however, 46% harbor intact *Tigger* elements, although the number of these intact elements is very low. Common HT events of *Tigger* elements were discovered across different lineages of animals, including mammals, that may have led to their widespread distribution, whereas *Helogale parvula* and arthropods may have aided *Tigger* HT incidences. The activity of *Tigger* seems to be low in the kingdom of animals, most copies were truncated in the mammal genomes and lost their transposition activity, and *Tigger* transposons only display signs of recent and current activities in a few species of animals. The findings suggest that the *Tigger* family is important in structuring mammal genomes.

## 1. Introduction

The movement of genetic material among reproductively separated species is known as horizontal transfer (HT). This form of transmission is ubiquitous in prokaryotes, where it is frequently used to generate genetic innovation [1,2]. HT is also becoming acknowledged as a major evolutionary driver altering eukaryotic genomes. So far, most HT events identified in eukaryotes are organelle-to-nucleus prokaryote-to-eukaryote gene transfers [3,4,5,6]. In fact, just a few occurrences of gene transfers across multicellular eukaryotes have been reported [7,8,9], while the vast majority of detected HTs within metazoans relate to transposable element (TE) transfers [10]. TEs are DNA pieces that can move across genomic regions, frequently replicating themselves throughout the process [11]. Due to their two distinct features, TE elements might serve as a means of interspecies gene transfer: they are capable of mobility, and they often represent the single most abundant component of eukaryotic genomes; for example, TEs make up 45% and 85% of the human and maize genomes, respectively [10,12]. 

One type of transposition mechanism of TEs is DNA-mediated, thus named DNA transposons (or class II transposons); most DNA transposons move using a cut-and-paste model facilitated by transposases. They are distinguished by the presence of terminal inverted repeats (TIRs) and target site duplication (TSD) [13]. DNA transposons are categorized into distinct superfamilies based on their transposases, including *pogo*, *Tc1/mariner*, *piggyBac*, *hAT*, *Zator*, *P*, *PIF/Harbinger*, *PHIS*, etc. [14,15,16].

The TEs, particularly DNA transposons, are the best-documented examples of HT between the nuclear genomes of multicellular eukaryotes [17,18]. Thus far, notable examples of HT of DNA transposons have been detected in diverse species such as insects [17,19,20,21], fish [22], nematodes [23], and plants in one case [24]. Retroviruses have invaded the germlines of various mammals [25,26,27,28], and there is accumulating evidence supporting the horizontal transmission of a snake retrotransposon in ruminants [29,30]. In summary, three criteria are employed to infer HT events: (1) significantly higher similarity of TEs, compared with non-mobile sequences; (2) non-congruent phylogeny between TE and host; and (3) spotty TE prevalence inside one group of taxa [17,31]. 

The *IS630-Tc1-mariner* (*ITm*) is one of the most common types of cut-and-paste DNA transposons. Elements of this group may be found in practically every part of the tree of life [32]. *ITm* transposons can be divided into different groups depending on the DDE/D signature. Recently, a novel *ITm* superfamily known as *Sailor* (DD82E) was discovered [33], which constitutes a unique superfamily with a discrete DDE domain (DD78-111E) and different evolutionary positions than prior superfamilies (*Tc1/mariner*, DD34E/*Gambol*, DDxD/*pogo*, *TP36*, and *Zator*) [34,35,36,37,38]. *Tc1/mariner*, a cut-and-paste transposon superfamily, is considered to be the most ubiquitous category of DNA transposons, with many different families, including *DD34D/mariner*, *DD37D/maT*, *DD39D/GT*, *DD41D/VS*, *DD34E/Tc1*, *DD35E/TR*, *DD36E/IC*, and *DD37E/TRT* [39,40,41,42,43,44,45,46]. 

The *pogo* element was first discovered in flies [47], followed by the discovery of a variety of related transposons, such as *Tigger* in bees and humans [48,49], *Aft1*, *Flipper*, *Pot1*, *Pot2*, *Tan1*, and *Fot* transposons in fungi [50,51,52,53,54,55], and *Lemi* in plants [56], which were phylogenetically close to *pogo* transposase [34,57,58]. For a long time, it was thought to be a member of the *Tc1/mariner* superfamily [34,35], and it was known as DD×D/*pogo* [34]. However, recent evolutionary analyses have revealed that *pogo* is a distinct superfamily [14]. Gao et al. (2020) found that *pogo*, *Gambol*, and *Tc1/mariner* constitute well-supported monophyletic clades, implying that they are distinct superfamilies that may have evolved separately from different clades of bacterial *IS630* TEs. They also highlighted that *pogo* transposons may have the broadest taxonomic distribution, compared with other *Tc1/mariner* and DNA transposon superfamilies [14]. *pogo* transposons were divided into six families: *Passer*, *Mover*, *Tigger*, *Fot/Fot-like*, *pogoR*, and *Lemi* [14]. The major *pogo* superfamily also includes a well-supported collection of numerous subclades [14]. 

In this paper, we analyzed the evolutionary connections, taxonomic distribution, and HT events of *Tigger* transposons in eukaryote genomes. The data we present show the complete evolutionary landscape of *Tigger* transposons and their invasion of different eukaryotes species genomes. These findings have major implications for determining the evolution of *Tigger* transposons and their influence on genome evolution.

## 2. Material and Methods

### 2.1. Mining of Tigger Transposons

*Tc1/mariner* transposons from the RepBase database were merged with reference sequences of *Tigger* transposons identified in previous research [14,48,59] to produce transposases sequences, to identify the distributions of *Tigger* transposons in the genomes. The *Tigger* transposase sequences were then used as queries in a TBlastN (with an E-value of 1^e-100^) search at the National Center for Biotechnology Information (NCBI) against the accessible organism genomes of prokaryotes and eukaryotes. The newly discovered sequences were then utilized as queries to find additional elements. The top 10 non-overlapping matches and 2 kb of flanking sequences were retrieved and manually aligned using the MAFFT v. 7.310 software [60] to determine transposon boundaries (TIR and TSD). Subsequently, a consensus or representative sequence of the discovered *Tigger* transposon was employed for further investigation. Following that, all obtained BLAST hits (with >1000 bp in length, >40% coverage, and >80% identity) were utilized to determine copy numbers. Furthermore, elements with many copies (>5) in the genomes were aligned using DAMBE to create consensus sequences [61].

### 2.2. Structure and Phylogenetic Analysis of Tigger

Protein domains were detected using the online hmmscan website’s profile hidden Markov processes (https://www.ebi.ac.uk/Tools/hmmer/search/hmmscan, accessed on 13 December 2020). Pairwise comparisons of full-length consensus or representative sequences were used to determine the sequence identities of *Tigger* transposons across species. To correctly identify the phylogenetic relationship of *Tigger* transposons, relevant open reading frame (ORF) sequences were obtained and translated into protein using Bioedit software. The sequences of the conserved DDE/DDD domains of *Tigger* transposases and other reference *pogo* families [14,59] were extracted and aligned with the MAFFT software [60]. Then, using the maximum likelihood approach in IQ-TREE [62], these DDE/DDD domain sequences were used to interpret the phylogenetic tree. ModelFinder integrated into IQ-TREE [63] chose the best-fit model, and the accuracy of maximum likelihood trees was evaluated using the ultrafast bootstrap technique with 1000 repetitions. 

### 2.3. Evidence of HT for Tigger across the Animal Kingdom 

Horizontal transfer events of *Tigger* transposons were detected using pairwise distances between the host genes and the transposons. To determine possible *Tigger* HT events, the pairwise distances between *Tigger* transposase-coding sequences and host gene-coding sequences were determined. Ribosomal protein genes are known to be globally conserved [64]. Therefore, we selected two highly conserved ribosomal proteins (RPL3 and RPL4) as the host genes, which also have a wide taxonomic prevalence in eukaryotes with one genomic copy, according to information in the orthologs database (OrthoDB), and were applied in our previous survey [33].

By introducing four critical filters, we employed a strict stand to avoid potentially false-positive estimates of *Tigger*’s HT events. The first approach was based on *Tigger* transposon sequence identity—only transposon sequence identity of pairwise species higher than 70% was selected for HT analysis. The second standard was based on the genetic distance of transposons between species: Only if the species had genetic distance 1.2 times smaller than host genes was kept for further HT analysis. Third, two host genes (RPL3 and RPL4) were employed for late HT deducing; HT events were recognized only if the genetic distance among species was significantly less (*p* < 0.01) for transposons than for all host genes. 

Two 60S ribosomal proteins RPL3 and RPL4 reported as universally conserved host genes [64] were analyzed for conservation and length and the taxonomic spread of the sole genomic copy among domains to pick the suitable host genes for the HT assumption analysis (Appendix A). The taxonomic distribution of host genes across eukaryote genomes was assessed using the OrthoDB web database (https://www.orthodb.org/, accessed on 25 December 2020). NCBI was used to retrieve all accessible gene annotations (coding sequences (CDS)) for host genes (RPL3 and RPL4) of species invaded by *Tigger*. The CDS of these genes was manually annotated by screening against the WGS via TBLASTN for those species whose host genes were not annotated. All acquired RPL3 and RPL4 sequences were collected, and sequences with considerable length deviations were excluded. Multiple alignments of RPL3 and RPL4 and *Tigger* were constructed, using the MAFFT software to detect HT events. Lastly, using MEGA (pairwise deletion, maximum composite likelihood) [65], the pairwise distances between the various species were computed for the *Tigger* and host gene (RPL3 and RPL4) coding sequences, respectively. 

Furthermore, regarding HT events of *Tigger* among the species, two additional genes (non-ribosomal gene) were used. Recombination-activating gene 1 (RAG1) [41,66] was used only for confirming the HT events in vertebrate species that were previously tested by RPL3 and RPL4 genes. However, the tubulin beta-3 (tub3) gene [67] was used for confirming the HT events in invertebrate species that were tested by using RPL3 and RPL4 genes. Statistical differences in genetic distances were examined using a one-factor ANOVA test in SPSS Statistics program v.25 (IBM Corp., Armonk, NY, USA).

## 3. Results 

### 3.1. Taxonomic Distribution and Structure Organization of Tigger 

The known *Tigger* transposase sequence [14,48,59] was applied as a query to explore against all available prokaryotic and eukaryotic genomes stored in the NCBI database in order to establish the taxonomic distribution of *Tigger* transposon. Overall, 383 *Tigger* homologous elements representing 325 species were collected (Appendix A) and submitted for phylogenetic analysis in the IQ-Tree software using the maximum-likelihood approach. Maximum-likelihood evaluation of the ORF protein sequences of the *Tigger* transposons, and known DNA transposases from *pogo* indicated that *Tigger* transposases constituted a major well-supported clade, with a 95% bootstrap value (Figure 1). Based on the phylogenetic tree, *Tigger* transposases were subsequently classified into five different intraclusters (Clusters A–E) (Figure 2). Cluster A had 12 vertebrate species (including Fishes, 2 species, and Reptiles, 10 species), and 60 invertebrates (Echinodermata, 1 species; Arthropod, 56 species; Platyhelminthes, 1 species; Mollusca, 1 species; Porifera, 1 species; and Cnidaria, 1 species). Cluster B had 48 vertebrates (Mammals, 33 species; Reptiles, 4 species; Fishes, 10 species; and Amphibians, 1 species), and 10 invertebrates (Arthropod, 5 species; Mollusca, 2 species; Platyhelminthes, 1 species; Annelida, 2 species; and Unrochordata, 1 species), while Cluster C had only 1 vertebrate (Reptiles, 1 species) and 23 invertebrates (Arthropod, 21 species; Nematoda, 1 species; and Platyhelminthes, 1 species). Moreover, Cluster D contained 18 vertebrates (Mammals, 7 species; Fishes, 3 species; and Reptiles, 8 species) and 22 invertebrates (Arthropod, 20 species, and Platyhelminthes, 2 species). However, Cluster E had only 1 arthropod species, and the other 60 species were mammals (Figure 2).

*Tigger* was first reported in humans, and also observed in invertebrates [48]. Our data revealed that the *Tigger* family’s distribution is restricted in animals but with extensive transmissions in both vertebrates and invertebrates; particularly, a wide expansion of *Tigger* was observed in mammals, where retrotransposons dominate the genomes, and DNA transposons are rare [68]. Overall, 169 *Tigger* elements were identified in 145 species from 29 orders of vertebrates, and 214 *Tigger* elements were found in 180 species from 36 orders of invertebrates (Figure 3 and Appendix A). *Tigger* was found in nearly all lineages of vertebrates; particularly, extensive expansion of *Tigger* was detected in mammals (103 species)—namely, Metatheria (3 species), Afrotheria (1 species), Carnivora (40 species), Cetartiodactyla (2 species), Chiroptera (14 species), Dermoptera (1 species), Insectivora (1 species), Lagomorpha (1 species), Perissodactyla (3 species), Pholidota (3 species), Primates (23 species), Rodents (9 species), and Xenarthra (2 species) (Figure 3 and Appendix A). Furthermore, *Tigger* was also found in Reptiles (20 species), Amphibians (3 species), and Birds (1 species). In comparison, in fish, which represents a great diversity of species and the major reservoir hosts of DNA transposons [69,70], only 20 species were found to harbor *Tigger* (Figure 3 and Appendix A). *Tigger* elements invaded nearly all invertebrate species as well, including Annelida (1 species of 1 order), Cnidaria (5 species of 2 orders), Cephalochordata (1 species of 1 order), Echinodermata (2 species of 1 order), Mollusca (8 species of 6 orders), Porifera (2 species of 2 order), Platyhelminthes (2 species of 1 order), Urochordata (1 species of 1 order), Nematoda (2 species of 1 order), and arthropod (155 species of 19 orders) (Appendix A). 

*Tigger* element copy numbers (>80% identity and 40% coverage) differed dramatically among these genomes, ranging from 1 to 2324 in each (Appendix A). In mammals and reptiles, most *Tigger* were short truncated copies, although some long copies were found; numerous long copies of *Tigger* were found in Cetartiodactyla, Perissodactyla, Fishes, and even invertebrates (Appendix A). More than the majority of the species (231/325) had at least one intact *Tigger* element, including 87% of Arthropod species (159/183), 83% of Fishes (15/18), and 45% of Mammals (45/120). In addition, intact *Tigger* elements were found in almost all vertebrate and invertebrate classes detected in this study (Figure 3 and Appendix A). The distribution of *Tigger* elements in different eukaryotic phyla revealed that this family is still active in these organisms.

The structure of the *pogo* protein, as well as the transposons, has been proven to be substantially conserved in previous research [14,49]. As shown in Figure 4, *Tigger*’s transposase consists of a CENP-B DNA-binding domain with a helix–turn–helix (HTH) motif at the N-terminus and a catalytic domain in the C-terminus (Figure 4) [14,48,49,71]. *Tigger* structure was discovered to be preserved throughout a variety of eukaryote species, including insects, fish, frogs, and bats. Most complete *Tigger* transposons were around 2.8 kb (range 1018–4525 bp) in length and contained a single ORF encoding a protein of about 492 aa (range 300–685 aa) flanked by short 20 bp (8–33 bp) TIRs (Figure 4). *Tigger* elements were discovered to be flanked by TA target site duplication (Table 1). The intact *Tigger* transposon in the Insecta *Cryptotermes secundus* is 2485 bp long, encoding a 537 aa transposase and flanked by 23 bp TIRs, and represents a common structure of this family (Figure 4).

### 3.2. Tigger Evolutionary Dynamics in Vertebrate Genomes

Our data revealed that the recent and current activities of *Tigger* tend to be low in vertebrates; only a few species contain high intact copies (around eight copies) of *Tigger*, such as *Dermochelys coriacea* and *Chiloscyllium punctatum* (Appendix A). Particularly in mammals, about 61% of those species contain more than 50 copies of *Tigger*; however, 46% harbor intact *Tigger* elements, but the copy number of these intact elements is low (Table 1 and Appendix A), indicating that the substantial expansion of the *Tigger* experienced in these species and mutation accumulation resulting in activity loss would occur in these copies. To further highlight the evolution patterns of *Tigger* elements in vertebrates, we used a Kimura divergence to evaluate the evolutionary dynamics of *Tigger* elements among the species containing more than three intact *Tigger* copies, the findings of which are described in Figure 5. Generally, young transposons represent low Kimura divergences [72], which reflects the activity of a transposon on a relative time scale per genome [73]. The data suggested the current activities of *Tigger* may only exist in one species of mammal (*Hipposideros armiger*) and one species of fish (*Latimeria chalumnae*), where some copies of *Tigger* displayed low Kimura divergences (<2%), while most *Tigger* copies in most species represent high Kimura divergences (more than 10%). This stated that *Tigger* old transposons invaded these species and may have lost their activities and become fossils (Figure 5). Some species (including *Euschistus heros*, *Clitarchus hookeri*, *Harmonia axyridis*, *Latrodectus hesperus*, *Girardia tigrina*, and *Schmidtea mediterranea*) experienced several waves of *Tigger* invasion, while several other species (such as *Colobus angolensis*, *Helogale parvula*, *Trichechus manatus*, and *Equus asinus*) underwent *Tiggers*’ solitary wave amplification. However, *Tigger* may still be active in some species of invertebrates— notably, arthropods and platyhelminthes, such as *Dysdera silvatica*, *Latrodectus Hesperus*, *Mesobuthus martensii*, *Girardia tigrina*, and *Schmidtea mediterranea*, where we discovered that many copies were intact in certain species with numerous *Tigger* copies (>5) and very high transposon sequence identities (Table 1 and Appendix A).

### 3.3. Evidence of HT Events for Tigger across Animals

The possible *Tigger* HT events were determined based on the strict stands described in Methods. Overall, 121 species involved in 13 HT events (Figure 6 and Appendix A) were recognized, with the genetic distance among species being significantly less (*p* < 0.01) for transposons than for the host genes (RPL3 and RPL4) (Figure 6A and Appendix A). Eventually, two non-ribosomal genes RAG1 and Tub3 were used to confirm *Tigger* HT events obtained from RPL3 and RPL4 ribosomal genes (Appendix A).

Using genetic distance comparison, *Tigger* HTs were detected in animals across many classes and orders. In detail, *Tigger* HT events were found in nine classes, with Eutheria being halfway between the other classes in *Tigger* HT events (Appendix A). In this study, most HT events were between species from the same phylum, specifically Chordata. However, some HT events were observed between different, including arthropods and Chordata (mammals, specifically Eutheria) (Appendix A). In detail, HT events of *Tigger* were detected between Eutheria (26 species) and Metatheria (2 species), Actinopterygii (7 species), Chondrichthyes (2 species), Agnatha (1 species), Sarcopterygii (1 species), Crocodilia (2 species), and Testudines (3 species), respectively (Appendix A). However, the HTs between Eutheria and arthropod (31 species) were the only between two different phyla detected in this study (Figure 6 and Figure 7A, and Appendix A). 

Moreover, HT events between fishes, specifically Agnatha (1 species *L. camtschaticum*) and Eutheria (26 species from 3 orders—namely, Carnivora, Chiroptera, and Rodents) were confirmed via phylogenetic relation, which is supported by Cluster B (Figure 2 and Figure 7B, Appendix A). This relation highlights the role of *L. camtschaticum* in highlighting *Tigger* transposon between Agnatha and Eutheria. Similarly, the HT events between Actinopterygii (seven species, including *Astatotilapia calliptera*, *Maylandia zebra*, *Oreochromis spilurus*, *Pundamilia nyererei*, *Takifugu flavidus*, *Takifugu rubripes*, and *Tropheus moorii*) and Carnivora (*Helogale parvula*) occurred across three clusters (Cluster B, Cluster E, and Cluster D) and *Helogale parvula* played the key role in those events. Additionally, *Helogale parvula* has played an important role in *Tigger* transposon HT between Carnivora and Sarcopterygii (one species, *Latimeria chalumnae*), as well as Chondrichthyes (two species—namely, *Carcharodon carcharias* and *Scyliorhinus torazame*), which were detected throughout three clusters (Cluster A, Cluster C, and Cluster E). We may infer that *Helogale parvula* was the essential factor of *Tigger* transposons HT events between Eutheria and Fishes (including Agnatha, Chondrichthyes, Actinopterygii, and Sarcopterygii) since it was the only Eutheria species that intermediated all HT events with fish. Additionally, HT events were detected between two orders of mammals including Eutheria (one species—*Helogale parvula*) and Metatheria (two species, including *Gymnobelideus leadbeateri* and *Trichosurus vulpecula*), which were confirmed phylogenetically in Cluster D and are summarized in Appendix A.

Furthermore, *H. parvula* also intermediated the HT events of *Tigger* transposons between carnivores (Eutheria) and arthropods (22 species, including *Drosophila athabasca*, *Drosophila azteca*, *Drosophila truncata*, *Dufourea novaeangliae*, *Delias pasithoe*, *Eufriesea mexicana*, *Glossosoma conforme*, *Hypothenemus hampei*, *Latrodectus hesperus*, *Marronus borbonicus*, *Megalopta genalis*, *Oryctes borbonicus*, *Osmia bicornis*, *Osmia lignaria*, *Schizaphis graminum*, *Scaptomyza flava*, *Tetragonula davenporti*, *Tetragonula hockingsi*, and *Tuta absoluta*), which were confirmed in three clusters (Cluster A, Cluster B, and Cluster C) (Appendix A). However, HT events were detected between Reptiles—namely, Testudines (three species, including *Malaclemys terrapin*, *Pelusios castaneus*, and *Pelodiscus sinensis*)—and Eutheria, specifically Carnivora (two species, including *Helogale parvula* and *M. lucifugus*), and those species belonged to different three clusters (Cluster B, and Cluster D, Cluster E) (Appendix A). Additionally, the HT event between reptiles, specifically Crocodilia (two species, including *Alligator sinensis* and *Gavialis gangeticus*) and Eutheria, specifically Carnivora (*Helogale parvula*) were confirmed phylogenetically and found to occur among three clusters (Cluster A, Cluster B, and Cluster E) (Appendix A). 

## 4. Discussion 

*Tigger*1 and *Tigger*2 were the first *Tigger* elements identified from mammalian genomes and defined in terms of their resemblance to *pogo* and CENP-B elements in general [47,74]. *Tigger*1 has been designated as a mammalian *pogo* [47]. Despite the fact that the transposase sequences of *Tigger*1 and *Tigger*2 are similar, they are very distantly linked in a larger evolutionary perspective [59]. *Tigger* elements are polyphyletic, meaning they do not belong to a single monophyletic group. Furthermore, the use of sequence similarity to categorize elements has resulted in the annotation of new *Tigger* elements throughout a wide range of *pogo* elements diversity over time. *Tigger* transposons belong to the pogo superfamily of transposons on the evolutionary tree. The analysis of the structural organization of the *Tigger* family enabled us to determine some notable distinctions between them and the members of the *pogo* superfamily. The most noticeable distinction was the DDE signature—the *Tigger* family was shown to include the DD29-36D catalytic domain [14,49,59]. 

In this analysis, we used the available TBLASTN and BLAST tools to scan the NCBI Whole-Genome Shotgun (WGS) library for *Tigger* transposons and generated their evolutionary profiles. Our findings show that *Tigger* transposons are broadly and unevenly dispersed in eukaryotes, invading 325 species across most invertebrate (excluding Ctenophora) and vertebrate groups (except for Caudata and Monotremata). Our research also found that arthropods and mammals are important reservoir lineages of *Tigger*, with 155 (19 orders) and 105 (in 12 orders) species invaded, respectively (Figure 2 and Figure 3). This prevalence of *Tigger* elements in mammalian genomes might suggest a distinct evolutionary profile of DNA transposons in mammals. However, the *Tigger* family’s taxonomic spread was underestimated due to the omission of shortened elements from older copies. Furthermore, mammals show distinct evolutionary profiles for the TE landscape than reptiles, amphibians, and fishes, with less diversity and activity of DNA transposons [70,75,76]. Despite the fact that certain DNA transposon families have invaded mammals’ genomes, many have restricted distribution, with just a few lineages entering mammalian genomes, such as DD41D/*VS* and DD36E/*IC*, two different *Tc1/mariner* transposon families [43,44]. Our findings indicate that *Tigger* represents a distinct phylogenetic landscape in mammals, with a broader expansion range than the other DNA transposon families investigated.

As TEs exhibit radically distinct evolutionary dynamics throughout vertebrate groups, reportedly active DNA transposons tend to be highly prevalent in Actinopterygii genomes than in Aves or mammalian genomes [36,70]. Although DNA transposons have invaded numerous mammalian lineages, most of them exist as incomplete copies in such genomes and have lost transposition activity, except *piggyBac* elements in Chiroptera, which have been characterized to have active copies [77]. In this study, the analysis of the evolutionary dynamics of *Tigger* elements revealed that mammalian species have high copies of *Tigger* elements, but their intact copies were low, which indicates that they appear as truncated copies in these genomes and have lost transposition activities. On the other hand, *Tigger* seems to be active in Arthropoda, Platyhelminthes, reptiles, and fish, which had high intact copies >5 in the genome, indicating that they are very young insertions in genomes, with most *Tigger* copies represented by very low Kimura divergences (Figure 5).

According to the current investigation, *Tigger* elements appear to be characterized by a high incidence of HT among animals. Uneven distribution of the *Tigger* transposon indicates the presence of putative HTs of *Tigger* elements in the animal genomes. *Tigger* transposons were found in numerous animal (vertebrates and invertebrates) lineages (mammals, fishes, reptiles, and arthropods) and displayed recurrent HT events (Figure 2 and Figure 3). HT events of *Tigger* between vertebrates and invertebrates were also observed (Figure 2, Figure 6 and Figure 7, and Appendix A). HT events of *Tigger* between vertebrates (Eutheria) and invertebrates (arthropods) were also observed (Figure 2, Figure 6 and Figure 7). 

Overall, HTs were confirmed by 124 species pairings depending on the genetic distance assessments among transposons and the 2 host genes. In addition, a higher incidence of HT was found in Eutheria, which may indicate why *Tigger* elements are more abundant in mammals. The *Tc1/mariner* and *pogo* superfamilies retain the highest for confirmed HT instances among TEs [71,78]. *Tigger* likely inherits the capacity to endure frequent HT. Many publications have been written after the discovery of HT, detailing this activity in numerous orders of animals, including arthropods, mammals, reptiles, etc. It has been demonstrated that these events can occur across lineage and distant taxa [71,79,80,81,82,83]. Many HT events have been identified to date, with approximately one-third of them being related to elements of the *Tc1/mariner* superfamily [78]. Considering the availability of information on HT transposons and the significant number of reported examples, the mechanism behind this phenomenon remains unknown. Issues regarding the likelihood of responsive insertions being generated in the recipient’s genome and their contribution to genome evolution and speciation remain unanswered. Furthermore, the discovery of new cases of HT will contribute to our understanding of the phenomenon and come closer to resolving the difficulties raised earlier. The detection of HT instances across eukaryotes illustrates the potential of genetic information being exchanged between two distinct species. 

Although most *Tigger* elements were retrieved from practically all animal host genomes, the host range typically differs across families. The processes underlying host range in TEs are poorly understood, and so are the trends we see for *Tigger* elements. For instance, if the host range is predominantly controlled via encounter or through compatibility criteria, the sensu hypothesis from host–parasite relations [84] is still an ongoing research subject. Our data displays a common horizontal transfer of *Tigger* elements across a wide range of hosts (Figure 6 and Figure 7). The most likely interpretation for the reported pattern is that each *Tigger* family was extant in the eukaryotic ancestor lineage and that active components representing each *Tigger* family were preferentially maintained in only a particular host lineage (Appendix A). Nevertheless, this would need inducing a significant range of loss events throughout the eukaryote evolution. As a result, given the improbability of the alternate explanation and recent studies illustrating the occurrence in which HT can emerge, we propose that a history of HT is the most probable answer for the identified host spread of *Tigger* elements. The current study of horizontal transmission in eukaryote genomes revealed that *Tigger* elements were the most commonly horizontally transferred. Despite their extremely careful analysis, the authors discovered multiple possible HT events involving *Tigger* elements between sixteen arthropod species (*Anopheles coluzzii*, *Anopheles merus*, *Cataglyphis niger*, *Dufourea novaeangliae*, *Eufriesea mexicana*, *Glossosoma conforme*, *Hypothenemus hampei*, *Latrodectus hesperus*, *Marronus borbonicus*, *Oryctes borbonicus*, *Osmia bicornis*, *Osmia lignaria*, *Schizaphis graminum*, *Sipha flava*, *Tetragonula davenporti*, and *Tetragonula hockingsi*) and one mammal species (*Helogale parvula*). In addition, we also discovered that all HT events for *Tigger* transposons occurred between mammals and all other animal lineages, emphasizing the importance of mammals in *Tigger* evolution. A further surprising feature of *Tigger*’s horizontally transferred transposons is the substantial participation of a mammal species (the common dwarf mongoose, *Helogale parvula*) in *Tigger*’s HT occurrences across eukaryotes classes, which may be relative to the specific ecosystem of this species involved, where the distribution of the common dwarf mongoose is very extensive and ranges from the East to southern Central Africa. At the same time, their diet is extremely diverse and consists of insects (mainly beetle larvae, termites, grasshoppers, and crickets), spiders, scorpions, small lizards, snakes, small birds, and rodents [85]. Therefore, we speculated that the wide distribution of *Helogale parvula* and diversification of its diets might facilitate HT events of *Tigger* across mammals.

*Tigger* displays a unique evolution dynamics in animal genomes, where this family represent low recent and current activities, and active copies are limited in a few animal species; most copies in mammals tend to be fossils, which is different from several well-defined families of *Tc1/mariner*, such as DD38D/*IT*, DD35E/*TR*, DD36E/*IC*, DD41D, and DD37E/*TRT* [36,39,41,43,44]. These families display relatively high recent and current activities in some vertebrate and invertebrate lineages, particularly in ray-finned fishes; multiple intact copies and low divergences across copies in many genomes were detected, indicating that they are young invaders and represent high current activities in this lineage, and some may be still active, for example, DD38E/*IT*, which has been proven to be able to transpose in human HeLa cells [36]. Although limited distribution (18 species) was observed for *Tigger* in fish genomes, the activity tended to be low, with few species containing fewer intact copies (<8), and only 2 vertebrate species (*Dermochelys coriacea* and *Chiloscyllium punctatum*) containing 8 intact copies of *Tigger*, indicating the overall activity of *Tigger* in animals is low.

## 5. Conclusions

This is the first study to thoroughly show the evolutionary profiles of *Tigger* transposons, which exhibit a very extensive taxonomic distribution in animals and have been horizontally transferred across diverse lineages of animals. However, low activities of *Tigger* were observed for most species. Importantly, we showed evidence that this family was extensively involved in mammal genomes’ evolution. This research adds to our knowledge of evolution, and its findings imply that the *Tigger* family plays an important role in shaping mammal genomes. 

## Figures and Tables

**Figure 1 biology-11-00921-f001:**
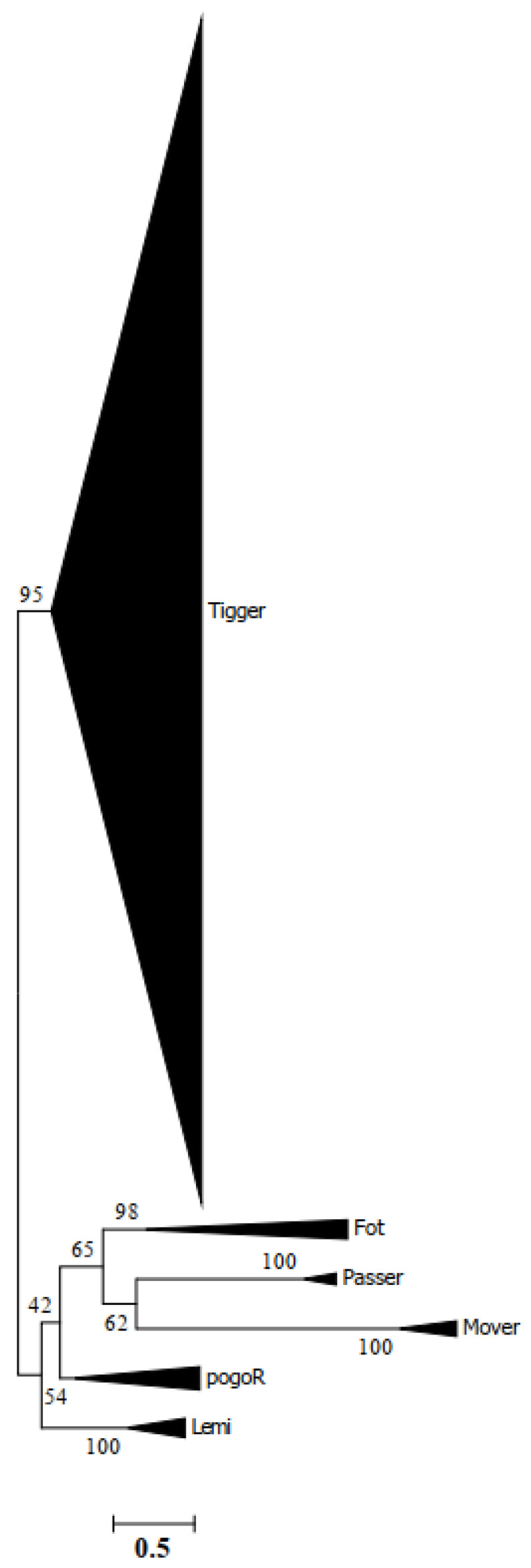
Phylogenetic tree of *Tigger* elements detected in this analysis with five other members of the *pogo* superfamily (*Passer*, *Mover*, *Fot/Fot-like*, *pogoR*, and *Lemi*) based on transposases. In IQ-TREE, bootstrapped (1000 repetitions) phylogenetic trees were inferred using the maximum-likelihood approach.

**Figure 2 biology-11-00921-f002:**
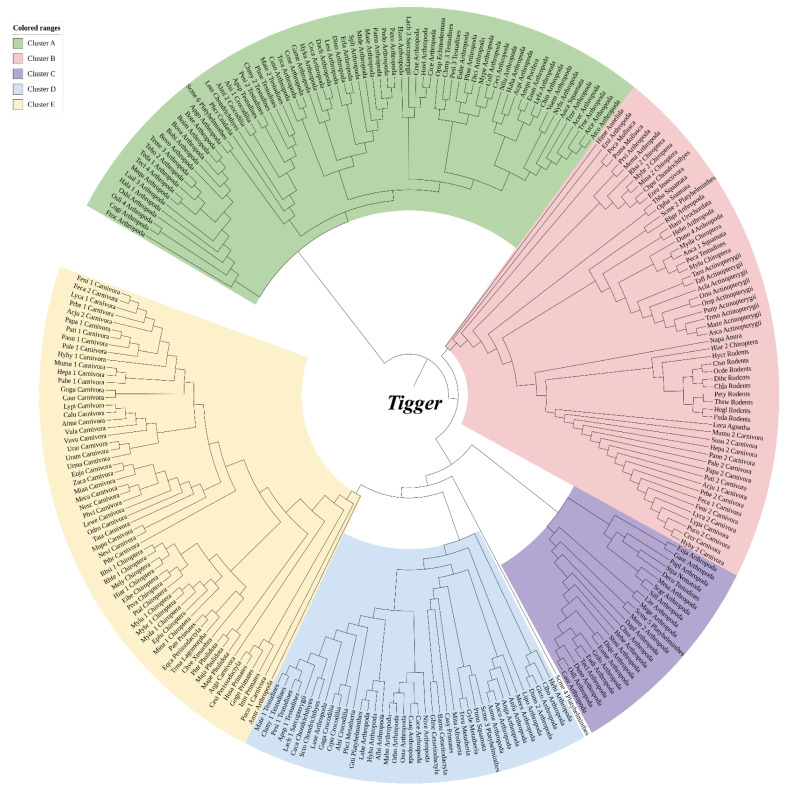
Intrafamily diversity of *Tigger* elements in the animal kingdom; each colored branch represents a phylogenetic tree cluster.

**Figure 3 biology-11-00921-f003:**
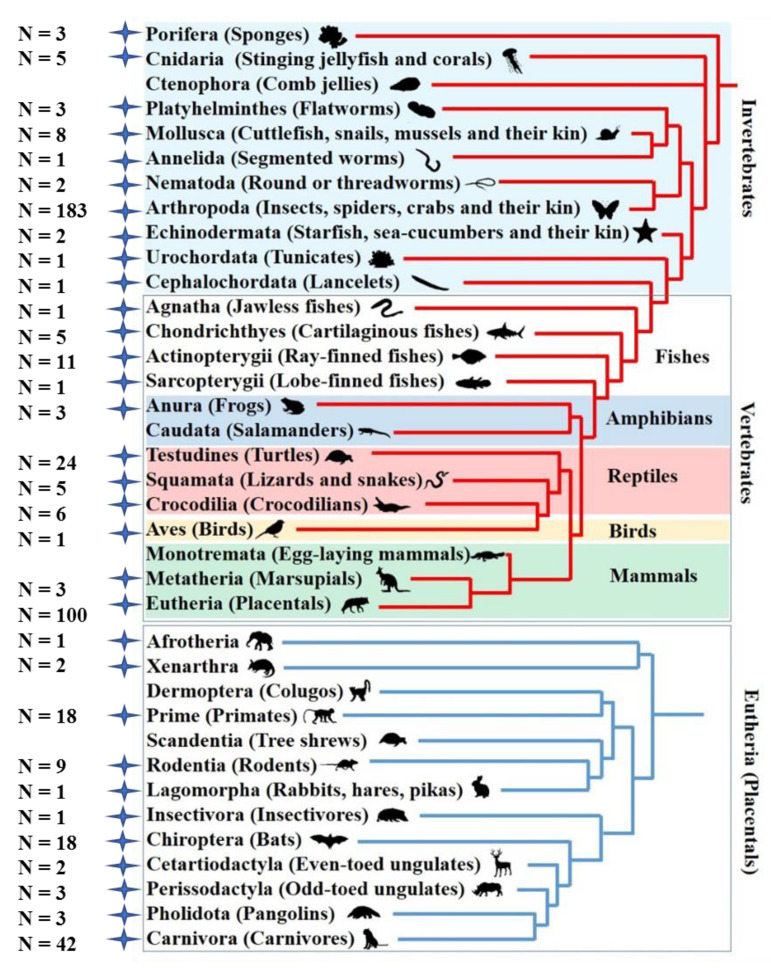
Distribution of *Tigger* transposons in eukaryotes. Blue stars show *Tigger* elements detected in the branches, and the numbers indicate the number of species possessing *Tigger* elements in each branch.

**Figure 4 biology-11-00921-f004:**
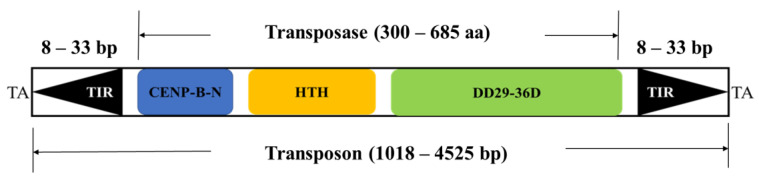
*Tigger*’s structural organization. TIRs are shown by black arrows, CENP-B-N motifs by blue rectangles, HTH sequences by yellow rectangles, catalytic domains (D: aspartic acid) by green rectangles, and TA is the target site of duplication.

**Figure 5 biology-11-00921-f005:**
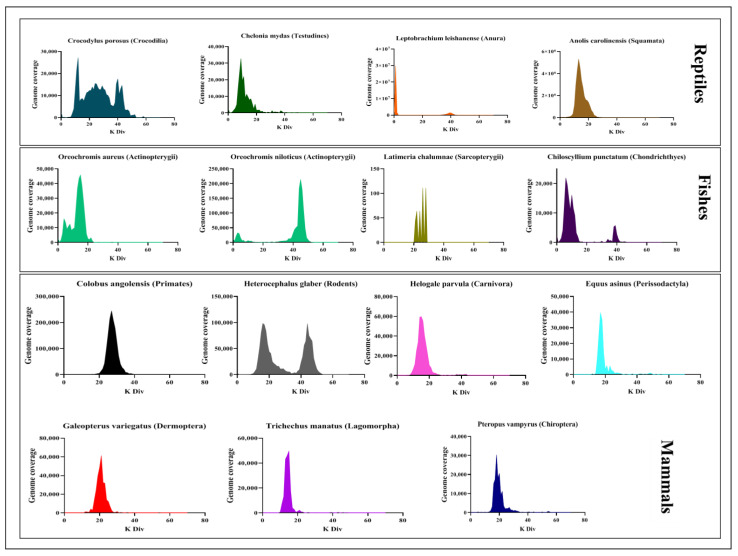
Evolutionary dynamics of *Tigger* in animals. RepeatMasker utility scripts were used to calculate the Kimura divergence from consensus sequences or the representative sequence (Tarailo-Graovac and Chen 2009). The *y*-axis represents the coverage (kb) of each *Tigger* element in the genome and the *x*-axis indicates the Kimura divergence estimate. Each color in the three orders (Mammal, Reptile, and Fish) represents species from different classes of the animal kingdom.

**Figure 6 biology-11-00921-f006:**
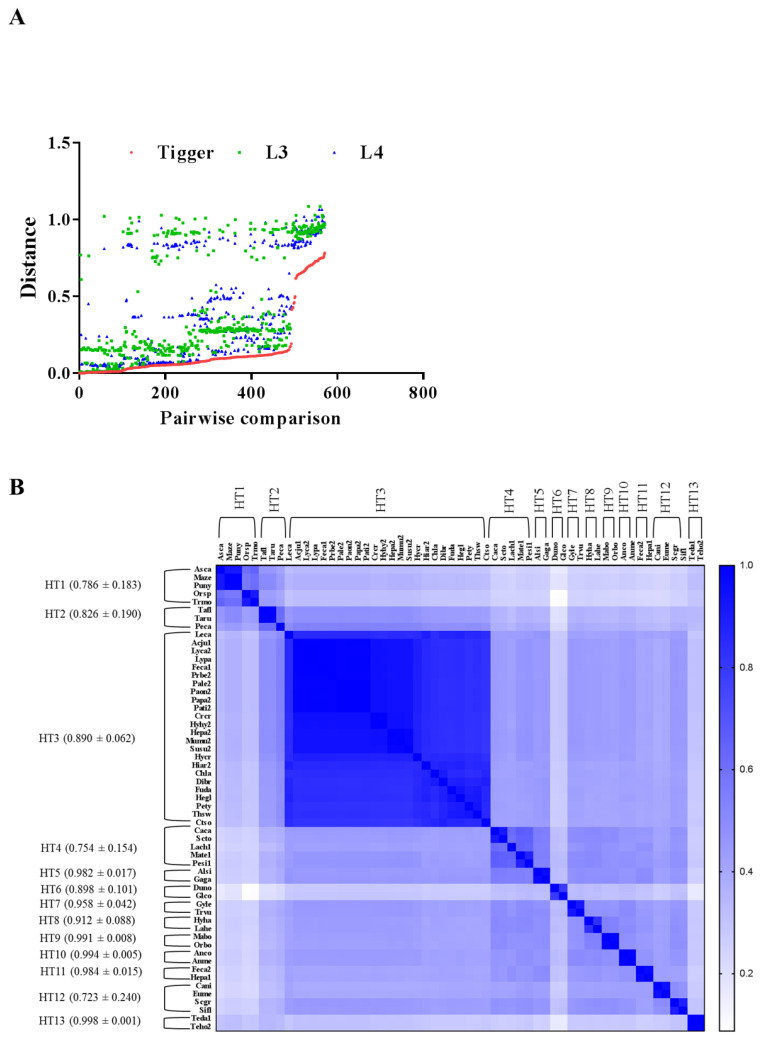
(**A**) Pairwise distance comparisons of *Tigger* transposons. The graph illustrates the pairwise distances of *Tiggers* and two organelle ribosomal proteins (L3 and L4) between the species included in this study, the red color represents *Tigger* transposon, the green color represents RPL3, and the blue color represents RPL4. The distances were obtained from all possible pairwise comparisons (N_L3_ = 572, N_L4_ = 572, labeled on the *x*-axis); (**B**) HT events measured among species contained Tigger elements. Sequence identities between *Tigger* elements among species. The sequence identities were measured via pairwise comparisons of *Tigger* CDS sequences (for species abbreviations, refer to Appendix A).

**Figure 7 biology-11-00921-f007:**
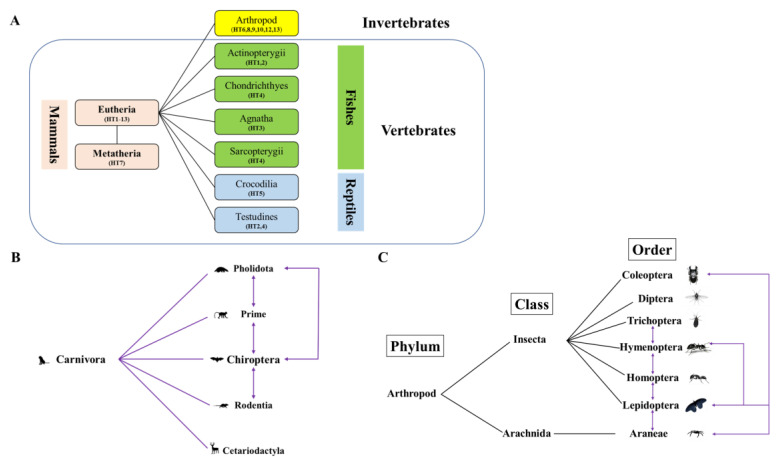
(**A**) The presented distribution and HT events of *Tigger* in eukaryotes; each black line represents HT events we detected, while the numbers represent the incidence of HT events of *Tigger*; (**B**) the HT events of *Tigger* within Eutheria; each purple line represents HT events we detected; (**C**) HT events of *Tigger* within arthropods. The black line represents Arthropod taxonomic classification; however, each purple line represents HT events we detected.

**Table 1 biology-11-00921-t001:** *Tigger* distribution in all eukaryotes examined in this study.

Distribution	Number of Species Containing *Tigger*	Number of Species Containing Full *Tigger* (%) ^a^	Number of Species Containing Intact *Tigger*	Length of the Full *Tigger* (bp) ^b^	Length of the Intact *Tigger* (bp) ^c^	Transposase Length of the Intact *Tigger*	TIR Length of the Intact *Tigger* (bp)	TSD
Porifera	3	2/66.6	2	2288–3872	2288–3872	334–685	20–741	TA
Cnidaria	5	4/80	2	1385–3309	1385–3210	374–538	8–24	TA
Platyhelminthes	3	3/100	3	2049–2762	2049–2762	360–598	17–25	TA
Mollusca	8	7/87.5	6	1609–3452	1609–2851	532–575	15–25	TA
Nematoda	2	1/50	1	2399–3400	3400	555	23–24	TA
Echinodermata	2	1/50	1	2988–3068	3068	540	21–23	TA
Urochordata	1	1/100	1	2381	2381	519	22	TA
Arthropods	183	159/87	147	1018–4225	1428–4225	301–607	12–33	TA
Annelida	1	1/100	1	1969	1969	396	24	TA
Cephalochordata	1	1/100	1	2872	2872	305	18	TA
Actinopterygii	11	10/91	9	1385–2821	1385–2821	301–438	13–23	TA
Agnatha	1	1/100	1	2906	2906	580	21	TA
Sarcopterygii	1	1/100	1	2099–2418	2099–2418	457–582	23	TA
Chondrichthyes	5	4/80	4	2094–4525	2094–4525	306–585	18–23	TA
Anura	3	2/66.6	2	1620–2277	1620–2277	340–597	13–29	TA
Squamata	5	4/80	4	2336–3988	2336–3988	532–618	17–26	TA
Crocodilia	6	3/50	3	2329–2516	2345–2360	587–640	11–24	TA
Aves	1	ND	ND	3808	ND	584	20	TA
Testudines	24	21/87.5	21	1370–2740	1370–2740	323–601	10–27	TA
Metatheria	3	3/100	3	2245–2379	2245–2379	541–546	20–24	TA
Eutheria	100	46/46	46	1605–2962	1605–2962	300–636	8–27	TA

^a^ The percentage of positive species. ^b^
*Tigger* elements flanked with TIRs were designated as full transposons. ^c^ The full transposons encoding intact transposases (>300 aa) were designated as intact transposons. ND: not detected; bp: base pair; TIR: terminal inverted repeats; TSD: target site duplication.

## Data Availability

All data needed to evaluate the conclusions in this paper are present either in the main text or in the Appendix A.

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
