# Peer review of "Revisiting the *Tigger* Transposon Evolution Revealing Extensive Involvement in the Shaping of Mammal Genomes"

_biology, 2022, doi:10.3390/biology11060921_

Round 1

Reviewer 1 Report

The manuscript describes the distribution of the Tigger transposon in species whose genome has been sequenced. While merely descriptive the manuscript try to explain the wide distribution of Toigger element through the inference of HT events.

There are several objective errors in the manuscript and the inference of HT events should be revised profoundly before the manuscript reaches the publication level. Moreover, the inference of HT should be performed in a more robust way to allow drawing conclusions, and a statistical evaluation should be presented.

Finally, there are many grammar errors and typos in the text. Therefore, I suggest the manuscript should be revised by a native English speaker. There are many more errrors in the text than I have indicated in my comments.

Please find below my specific comments.

Line 41-42 please revise this sentence for punctuation and grammar

Line 45 "One type TEs was DNA-mediated..." Do you mean one type of transposition mechanism? Also use the present form when referring to the currently accepted mechanism of transposition

line 53. Also in Nematodes there are documented examples of DNA transposon HT (https://doi.org/10.1016/j.ympev.2021.107090)

line 57 "...significant sequence similarity of TEs..." I would say "significantly higher similarity of TEs compared to non-mobile sequences.

line  86 "...their transfection..." transfection? please revise

line 97 remove an extra dash between newly and discovered

line 100 "encoded" should be encoding

line 122 "...the evolutionary relationship of the domesticated proteins..." I do not get the point here. What kind of domesticated proteins are the authors talking about?

line 128 "ribosomal protein is known to be a globally conserved gene" This is nonsense. Please revise

Methods: Are the thresholds used in the HT inference analyses arbitrarily chosen or were they obtained from the current literature? Please state clearly what is the rationale for using these parameters or cite the references.

Methods: Using two ribosomal proteins as reference for the inference of HT events could be as a double edge sword. While they are conserved proteins and it can be easy to find the corresponding genes in the DNA databases, this could represent a significant bias of the inference. Moreover we don't know if these genes are also horizontally transferred. I suggest to introduce more genes in the reference set, not only those encoding ribosomal proteins.

In the method section (or at least at the beginning of the results) the authors should state why they have

Paragraph 1.1 (Results)

line 160 "Tigger homology elements".....homologous?

line 198 "A substantial expansion.." could be this due to the high reptile sampling in the database (if compared to amphibians), rather than a real expansion in this taxonomic Class?

line 199 "...but few species in amphibians..." are amphibians reptiles too?

line 223 "Tigger's transposase consists of a CENP-B protein..." are you sure?

line 230 Since not all the readers are familiar with the taxonomic names of the species mentioned ion this manuscript the authors should consider inserting the type of organism they are talking about (e. g. ... in the ant C. secundus). Also, the complete taxonomic name (e. g. Cryptotermes secundus)

Table 1 footnotes. It is not clear if the a full-length element has BOTH TIRs and intact transposase gene, or if elements with EITHER TIRs or transposase genes are regarded as full-length.

Line 243 "the recent and current activities of Tigger" Current activity has not been demonstrated in this manuscript. It requires transposition assays to demonstrate transposition. The authors refer several times to the transposition activity of Tigger in the text but without experimental poof

The main issue with this manuscript is the lack of robustness of HT. The analysis was made using a pair of ribosomal proteins. the authors state that "sequences with considerable length deviations were excluded" (line 151). Which were the exclusion parameters? This exclusion introduce a bias in the analyses. Did the authors found intron-containing genes? If yes, how were the exon-intron junction predicted/recognized? Finally, a statistical estimation of the results is missing, which is essential to support the conclusions drawn. I suggest to reconsider the HT inference following my suggestions.

Figure 6 is difficult to read. Species are abbreviated in a weird way and it is not easy for the reader to understand if the grouped species are close or distant relatives

It is possible to speculate on the role of known or putative vectors that disseminate Tigger among species?

Reviewer 2 Report

This is an exhaustive work to characterize tigger elements in the widest range of living species.  But, there are issues that need resolution.  The biggest problem readers will have regards seemingly contradictory statements and conclusions regarding Tigger elements in animal genomes.  For example, the last sentence of the Summary says "the activity of Tigger transposons tend to 17 be low in vertebrates, only one mammalian genome and fish genome may harbor active Tigger." Yet then immediately following  in the Abstract: "Tigger was found in a broad variety of animals, including 180 19 species from 36 orders of invertebrates and 145 species from 29 orders of vertebrates, particularly, 20 extensive invasion of Tigger was observed for mammals, with 46% detected species genomes har-21 boring Tigger elements and 61% species contain more than 50 copies of Tigger. " and a bit later, "The activity of Tigger seems to be low in the kingdom of animals" Later, (lines 243-245) "Particularly in mammals, about 46% of mammal species contained intact Tigger elements, however, very few species harbor more than 3 intact copies of Tigger, many species (61%) have high copy numbers (more than 50 copies).  This confusion is pervasive. I think it is because the authors fail in the beginning to distinguish between autonomous transposons (those that have an active transposase gene and intact ITRs and hence can transpose themselves and, on occasion, non-autonomous transposons that have inactive transposase genes but still can be mobilized by transposase from autonomous units.  The authors are encouraged to construct an early figure that distinguishes these two types of elements and keep that idea alive throughout the report. For instance, this can explain the sporadic bursts of transposon numbers in some select genomes.

Once this aspect is clarified, then the discussion should be re-written in terms of the effects of autonomous transposons on their own spread as well as mobilizing non-autonomous transposons. This may then clarify what the last sentence of the paper means because just characterizing the tigger populations in many species does not indicate their playing a significant role in shaping genomes.

Smaller issues:

19: animals -> animal genomes

45: One type of TE is 

46: cut-and-paste

94: It's not clear what this number of significance refers to....

128: Ribosomal protein genes and their encoded proteins are ....

233: Define abbreviations used in the table 

271: Define colors used in the figure

281: Define color scheme in Panel A

434: It's not clear what this sentence means

Round 2

Reviewer 1 Report

The authors have responded to many of the questions raised in my previous report. However I have still concerns that the HTT inference has been weakly performed.

The authors haven't increased the number of reference genes in the HTT inference analysis, as suggested. As I stated in my first report, I believe that this is introduce a great bias in the analysis. The authors state in their rebuttal that they "also believe that adding more genes for genetic distance comparison, will not change the conclusion". Well, this is absolutely self-referencing and they need to demonstrate their statement.

Can the authors, at least, improve the analysis providing an increased gene number for the HT events depicted in supplementary file S2? Using additional reference (non-ribosomal) genes will both enforce the strategy of using two Ribosomal genes as reference and will also provide stronger evidence for the detected HTT.

 If the authors do not want to utilize more genes as reference they should highlight in the discussion that this represents a weakness of their analysis and they should tone down the discussion of the HTT finding.  

Reviewer 2 Report

Suggestions 

51: piggyBac

59: non-congruent phylogeny

68: family is considered

123: ribosomal protein-encoding genes (or ribosomal protein genes)

132: The sentence "Only if the species ..." does is not clear to me; what does 1.2 times smaller mean (the number- how is it calculated?

137: Here and especially later the huge use of parentheses is not needed (and is distracting); in this case leave them out and the sentence makes perfect sense

146: RPL4 sequences

174: Simplify - Cluster E had only one arthropod species and the other 60 species were mammals.

177: elements detected

181: kingdom; each 

187: the words "very" (used here), high, low, etc. have no scientific meaning and should be omitted - it will not affect meaning or clarity

195: Here and elsewhere, to be consistent, capitalize Supplimentary Table (or Figure)

198: only 20 species harbor (or "have") Tigger

204: Supplimentary Table

217: what is an "early-age clade; could simplify by just saying "...reveals this family is still active..."

221: DNA-binding

245: what does "very low" mean?

278, 288, 332: Tables S2 & S3

344: Omit "probably"

360: mammalian

252: ....confirmed current activities... (low and recent are without context)

372 & 375: 324 is average, 1.1 is low and 5 is high ????- this is why using such numbers is confusing without giving a context for these adjectives

375: ...fish, which have more than 5 intact copies/genome(?), indicating that Tigger invaded these species within the past XXX years (define what you ea by "recent age")

456: ....profiles of Tigger...

461: I think most readers will not see any important role for Tigger from this paper; it's a parasite that if it does NOT alter the phenotype, it is passed on until its activity decays and so it will have little consequence. There are NO examples in this paper of Tigger making any difference to those organisms that harbor active or inactive elements. Their presence is pretty insignificant overall, which is why they are tolerated

Round 3

Reviewer 1 Report

I appreciate the authors' effort in improving the manuscript following my suggestions.

My last suggestion is to carefully revise the text for residual hard-to read sentences and typos.